# Research on Visual Preference of Chinese Courthouse Architecture Appearance

**Jingyin Pan, Yuping Yuan, Xinyu Wang and Chenping Han ***

School of Architecture and Design, China University of Mining and Technology, Xuzhou 221116, China; panjingyin77@126.com (J.P.); 02140660@cumt.edu.cn (Y.Y.); wangxinyu605a@163.com (X.W.)
* Correspondence: hanchenping@cumt.edu.cn; Tel.: +86-151-6224-7027

**Abstract:** Whether courthouse architecture should be solemn or friendly is a topic worth exploring. Currently, a large quantity of courthouse buildings have been constructed in China, but the architectural style differs greatly. This situation provides a good foundation for the study on visual preference appraisal of the external form of courthouse buildings. This research selected 50 representative courthouse buildings as the research object and set architectural style, height–width ratio, window–wall ratio, and open ground in front of the buildings as the physical properties under study. Participants of different demographic characteristics were chosen to conduct a photo stimulation experiment. The results obtained were analyzed by SPSS. According to the research results, the neoclassical courthouse building with a large window–wall ratio, small height–width ratio, and large open ground is the most favorite among the participants; participants of different demographic characteristics render different visual preference appraisals of the external form of Chinese courthouse buildings. Contemporary Chinese court building should have a new image. This research can be taken as a reference for the external design of Chinese courthouse buildings.

**Keywords:** courthouse architecture; demographic characteristics; external space form; visual preference appraisal

## 1. Introduction

In the draft of 'Making Federal Buildings Beautiful Again', former US president Donald Trump demanded that all the office buildings of the U.S. Federal Government should be designed to be of classic style. Although his proposal stirred up tremendous disputes among the public, it also reveals partially that the external space form of buildings such as courthouses is believed to be solemn and uniform. In the book "the Fountainhead", written by American writer Ayn Rand, the protagonist of the novel, Howard Locke, was expelled from the Staten Institute of architecture for refusing to follow the stale and outdated traditions of the school. Later, he designed a temple dedicated to the human spirit, but it violated the design requirements of customers. This novel discusses the contradiction and connection between people's visual preference and the architecture's own aesthetics by using the character image of the talented architect Locke [1].

In addition, some researchers have also maintained that compared with the traditional courthouse buildings, the modern courthouse buildings are highly impressive in terms of size, building materials, and color, and bring forth a stronger sense of coercion among the group who are familiar with courthouse buildings [2]. The authority and complex legal system brought about by the court building will bring people a sense of confusion. People hope to get rid of legal confusion and create legal rationality [3]. The torture of prisoners was gradually replaced by the prison regulations controlling them, as well as a series of related political, legal, power-related, conceptual, scientific and technological problems. Foucault M believed that rule gradually changed into hidden and psychological rule, and that when people face the courts and other judicial organs, their psychology and behavior will change [4].

### 1.1. Courthouse Architecture

Courthouse architecture is an important component of national justice and serves as the symbol of justice among the public [5]. The implication of courthouse architecture is to arouse people's belief in judiciary and pursuit for justice through distinctive style and images. The glass facade of courthouse architecture can be interpreted as the accessibility and transparency of legal justice [6]. Courthouse architecture should be a window through which the public can have a glimpse of culture and ideology and, thus, contribute to the establishment of fundamental political framework for the future society [7].

In ancient China, the style and form of courthouse architectures were relatively uniform. The courthouse architectures are constructed in the form of yamen (government office in feudal China). As the symbol of judiciary and power, yamen reflected the unification of administrative function and judicial function and served as the main place where local governors exercised governance and judicial powers [8]. In modern China, with the exchange of traditional ritual ideology and Western culture, the style of Chinese courthouse architecture became diverse. To be specific, Western classical style, modernism style, postmodernism style, and the combination of Chinese and Western styles appeared in Chinese courthouse architectures [9]. To sum up, the modern Chinese courthouse architectures tend to be diverse in style.

### 1.2. Visual Preference Appraisal

Visual preference appraisal is a widely used method to assess the influence of different architectural elements on people's visual preference. Visual preference is caused by human activities. On the other hand, visual quality is the result of the interaction between the vertical and horizontal perspective of an image object [10]. In this sense, the visual effect can be described from a quantitative angle [11]. Visual preference appraisal is mainly divided into two approaches: the objective, based on a physical paradigm, and the subjective, based on a psychological paradigm. The objective approach regards aesthetic quality as an intrinsic attribute of a building [12]. As a device that can measure intentional and instinctive visual responses, eye trackers may become a solution to study visual preference evaluation [13]. Sussman A and Ward J explored how government buildings affect human perception through eye movement experiments on Boston City Hall, and expanded the application scope of visual preference evaluation in the public domain [14].

The significance of visual preference appraisal lies in that it can replace the study of individual psychology or psychological research [15]. In other words, the focus of visual preference appraisal is not placed upon individual mentality but common significance of this phenomenon. Visual quality is the most important characteristic that directly influences the public's visual preference for architecture [16]. Although a photo display has certain limitations [17–19], it is still the most widely used and the most effective method in terms of aesthetic assessment [17,20]. In previous research, photos were widely used to replace the actual landscapes [21] and buildings [22,23]. Moreover, scholars also studied the influence of physical attributes on landscape aesthetics and managed to establish the correlation between physical attributes and landscape aesthetics [24,25]. Visual preference appraisal has emerged as a means of engaging the public in the public infrastructure planning and design process and providing public advisory resolution for government departments [26].

### 1.3. Demographic Characteristics

As has been revealed by previous research, people's visual preference appraisal of buildings would be influenced by many factors, such as the physical factors of buildings and the landscape factors around the buildings. Besides the above-mentioned factors, demographic characteristics of the interviewees are also highly influential [22]. Among the demographic characteristics, culture background [27], education level [28], gender [29,30], age [31], professional knowledge [30,31], familiarity with the environment [32], and living environment are all influential for people's visual preference appraisal [27,33].

In this paper, three characteristics are considered, namely, age, gender, and education level. These are the most recurring in studies on people's visual preference appraisal [34]. The working staff of a courthouse, the architects who design the courthouse buildings, and urban pedestrians walking by the courthouse all interact with courthouse buildings in different ways; thus, they will have different feelings about the structure [35]. Therefore, we did not limit specific groups when issuing the questionnaire.

### 1.4. Research Basis

Architectural style could influence people's visual preference for buildings [36]. The external appearance and image of courthouse buildings exert an imperceptible and unobtrusive influence upon the public's feelings. The difference in the external forms of courthouse buildings is highly influential to the public's appraisal. An experiment conducted in Malaysia indicated that the public have become increasingly interested in buildings of a modernist style [37]. A study on Fox Art Architecture of the late 19th century revealed that courthouse architectures of this style were rich in visual images, which conveyed messages about the judiciary process and society [38].

The complexity of building facades also influences people's visual preference [39]. To be specific, the complexity of facades is related to windows, whose spaciousness and transparency exert a certain impact on people's visual preference for buildings [40].

The height–width ratio of buildings also influences people's visual preference. The visual impact of high-rise buildings upon the viewers may be weakened through adjusting the building's height and harmonizing the building with its surroundings [41]. Some scholars have used an Augmented Reality (AR) method to study the harmonization of buildings of different heights with their surroundings. In this method, a number of 3D virtual rectangular objects with a scale are located on the grid of 3D geographical model. Then, the Computer Graphic (CG) models are displayed in an overlapping manner with the actual landscape from multiple viewpoints using the AR technology. The user measures the maximum invisible height for each rectangular object at a grid point [42]. As has been revealed by some studies, the height, roof, and color of high-rise buildings are the most important physical characteristics [43]; among the three, the height is the most influential factor for people's visual preference appraisal of high-rise buildings. Moreover, other studies also indicated that the change of building volume in a landscape could lead to significant differences in the public's visual preference appraisal [44]. Accordingly, the height–width ratio of buildings is taken as one variable in this paper.

The natural environment around buildings has some impact on people's visual preference as well. Natural landscapes are positively influential to human beings [28]. Similarly, whether there is greening landscape around the building also influences people's visual preference for the building [45]. The courthouse buildings should have their own natural environment, which is of help to distinguish it from the surrounding buildings [46].

In this paper, through a literature review, the building characteristics are summed up, and four of them are selected as the variables under study. Namely, they are architectural style, height–width ratio, window–wall ratio, and open ground in front of buildings. The method of summing up the characteristics of research objects through a literature review is widely used in research papers [12,22].

Currently, the architectural style of Chinese courthouse can be roughly classified into four types: modernist style, Chinese style, neoclassical style, and post-modernist style. These four styles differ from each other significantly, which makes it necessary to study these styles by means of classification.

### 1.5. Research Questions

When investigating people's opinion or satisfaction degree on public buildings, researchers tended to use the method of POE (short for Post Occupancy Evaluation) [47]. To be specific, this method focuses on how the environment embodies and meets the demands people express explicitly or implicitly. When using POE, researchers tend to consider more

upon building occupants' satisfaction with the internal space of buildings, but neglect the visual preference appraisal of the external space of buildings [48]. Accordingly, this paper also uses photos to replace the actual buildings so as to divert people's focus to the external space form of Chinese courthouse buildings.

With the external space form of Chinese courthouse buildings as a research object, this paper aims to explore the following questions:

- What kind of external space form is more appropriate for Chinese courthouse buildings?
- What physical factors would influence people's visual preference appraisal of the external space form of courthouse buildings?
- Do people of different demographic characteristics differ from each other in their visual preferences?

During the experiment, the data on participants' visual preference appraisal of courthouse buildings and the demographic characteristics of participants were collected by means of a questionnaire survey. Through analyzing and computing the data and demographic characteristics collected, the following two topics were studied: whether people with different demographic characteristics would render different visual preference appraisal of the external space form of courthouse buildings and whether different demographic characteristics cause a shift of focus on the physical properties of courthouse buildings.

## 2. Research Method

### 2.1. Research Site

Given the vast territory of China and the consequent significant cultural differences, courthouse buildings in China differ from region to region in their architectural style. Accordingly, when selecting research targets, this study selected fifty courthouse buildings distributed in ten cities, which are located in East China, South China, North China, Central China, Northeast China, Southwest China, and Southwest China.

### 2.2. The Calculation of Buildings' Physical Properties

2.2.1. Determining the Architectural Style of the Courthouse Buildings

The architectural styles of the fifty courthouse buildings were counted and the top three styles were chosen as the research materials in the end (see Table 1). The three styles were the modernist style, neoclassical style, and post-modernist style. The modernist style emphasizes the coordination between architectural form and function, simple modeling lines, and the use of industrialized materials, the neoclassical style advocates using modern materials and processing technology to pursue the general outline characteristics of the traditional style and simplify the classical column type and the post-modernist style pays attention to the historical context, and the architectural language has the characteristics of metaphor, symbolism, and polysemy.

**Table 1.** The statistics of architectural styles of Chinese courthouse buildings.

| Architectural Style | Number | Proportion |
|---|---|---|
| Modernist style | 16 | 32.00% |
| Post-modernist style | 11 | 22.00% |
| Neoclassical style | 23 | 46.00% |

When the three architectural styles to be studied were determined, the actual architectural forms of each style need to be set as well. The photos of the fifty courthouse buildings taken before were sent to five experts in architecture. They were asked to pick out the photos which could best represent the architectural style of courthouse buildings. For each architectural style, three photos were singled out. The top three photos that had been singled out by the five experts were chosen to represent the actual architecture forms of each style. Given the differences in shooting angle and light and shade, the nine

chosen photos were modeled by means of SketchUp2018, the building facade in each photo was selected to model in equal scale, retaining the building main body and enclosure, and simplifying the surrounding building environment. Then, the model was imported into Lumion 8.5 for rendering. According to Report on Nutrition and Chronic Disease in Chinese residents released by the State Council Information Office in 2019, the average height of adult males over 18 years old is 167.1 cm. accordingly, the shooting angle was set at 167 cm from the ground when the photos were modeled and rendered. The synoptic background was uniformly set as a cloudless sky. In the end, nine photos were obtained, as shown in Figure 1.

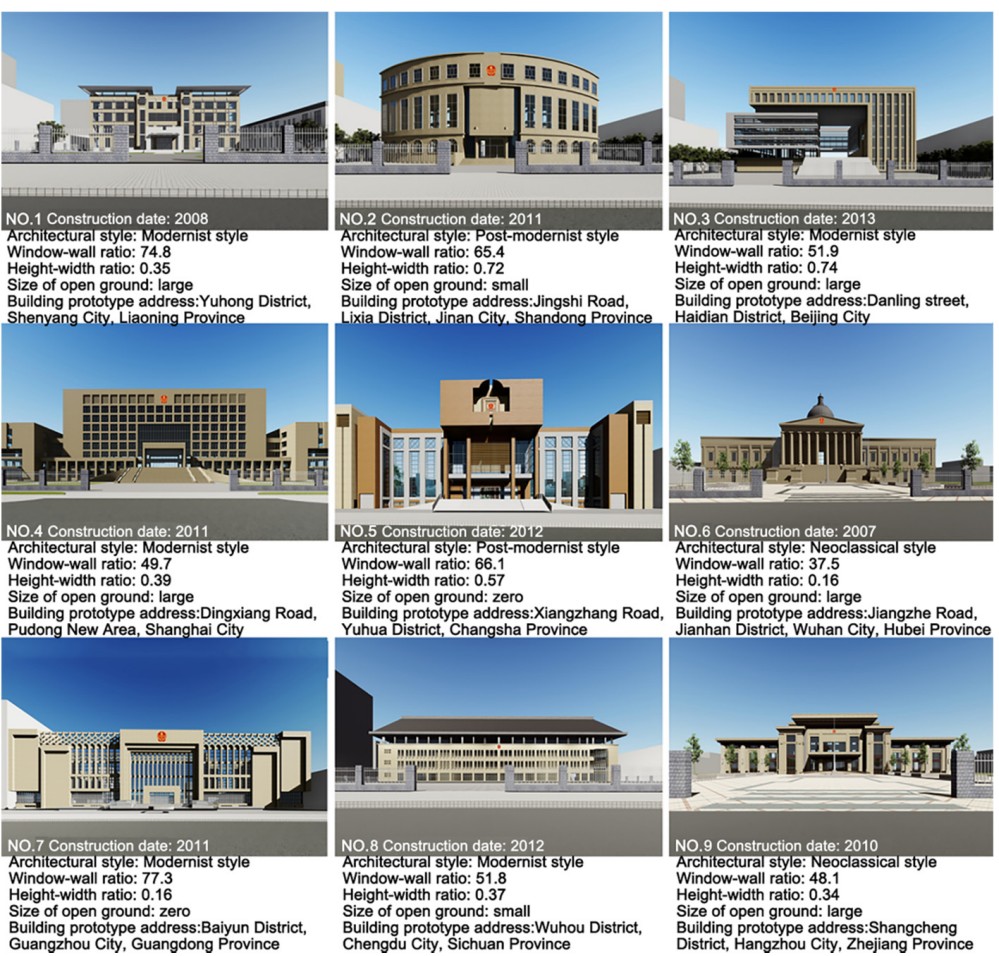

**Figure 1.** The nine photos of the Chinese courthouse buildings.

### 2.2.2. Calculating the Height–Width Ratio of the Courthouse Buildings

The change of building height and width influences participants' overall visual preference appraisals to a large degree. Therefore, in this experiment, the front facade of the courthouse buildings was taken uniformly to calculate the height–width ratio, as shown in Figure 2. The change of height and width is expressed by the height–width ratio. In other words, the ratio = the height of the front facade (the length of A)/the width of the front facade (the length of B). The calculation results are shown in Table 2.

### 2.2.3. Calculating the Window–Wall Ratio of the Courthouse Buildings

The relative area of windows in the courthouse buildings can be expressed by the window–wall ratio. To be specific, a = S1/S2, where S1 denotes the area of window in the facade and S2 represents the area of the facade. The front facade of the courthouse buildings was taken uniformly to calculate the window–wall ratio. The calculation results are shown in Table 3.

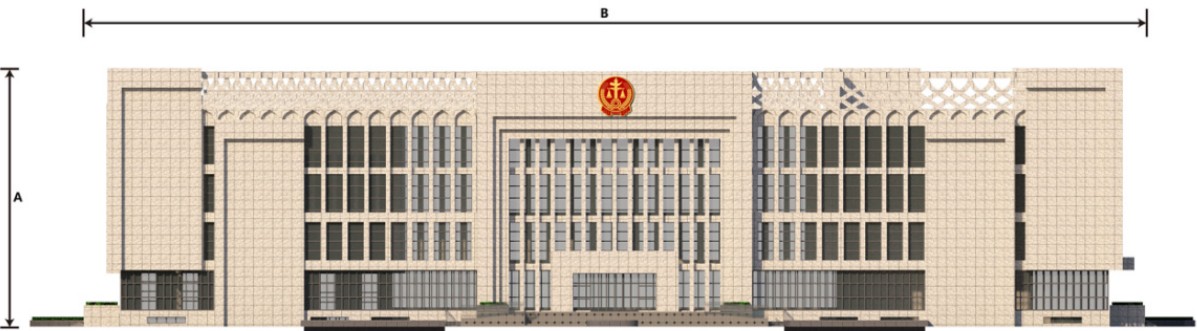

**Figure 2.** Schematic diagram of the height–width ratio calculation.

**Table 2.** The statistics of the height–width ratio of the Chinese courthouse buildings.

| Photo No. | The Length of A (m) | The Length of B (m) | Height–Width Ratio |
|---|---|---|---|
| No. 1 | 11.1 | 32.1 | 0.35 |
| No. 2 | 19.2 | 26.7 | 0.72 |
| No. 3 | 18.3 | 24.6 | 0.74 |
| No. 4 | 26.7 | 69.3 | 0.39 |
| No. 5 | 24.6 | 42.9 | 0.57 |
| No. 6 | 12.6 | 78.3 | 0.16 |
| No. 7 | 10.5 | 63.6 | 0.16 |
| No. 8 | 17.7 | 47.4 | 0.37 |
| No. 9 | 18.9 | 54.9 | 0.34 |

**Table 3.** The statistics of the window–wall ratio of the Chinese courthouse buildings.

| Photo No. | Mean Window–Wall Ratio |
|---|---|
| No. 1 | 74.8 |
| No. 2 | 65.4 |
| No. 3 | 51.9 |
| No. 4 | 49.7 |
| No. 5 | 66.1 |
| No. 6 | 37.5 |
| No. 7 | 77.3 |
| No. 8 | 51.8 |
| No. 9 | 48.1 |

2.2.4. Calculating the Size of the Open Ground in Front of the Courthouse Buildings

Given that the types and forms of the open ground in front of courthouse buildings were relatively complicated, the open ground was simply classified into three categories in terms of size: large, small, and zero. To be specific, firstly, the mean value of all the open grounds in this experiment was calculated; then, those larger than the mean value were defined as "large", while those smaller than the mean value were defined as "small". If there was no open ground in front of the courthouse building, it was defined as "zero", as shown in Figure 3.

*2.3. The Preference Survey of Participants*

In order to avoid special feelings that urban residents may have about the courthouse buildings in their city, we chose Xuzhou, Nanjing, and Zhengzhou to conduct the questionnaire survey. The nine as-prepared photos were shown to the participants. As has been stated above, the nine photos embodied different architectural styles, height–width ratios, window–wall ratios, and open ground in front of the building. For the convenience of the participants, they were printed on three pieces of full-color A4 photo paper, three in one piece, and bound in a volume in a random order. Then, these photos were shown to people

passing by who were asked to score the photos. The survey was conducted at weekends to minimize the possibility that the demographic variables were too homogenized. Moreover, the survey avoided the sites where the courthouse buildings in the photos were located. In the previous relevant studies, photos were widely used to replace actual landscapes [49], which justifies the practice of using photos to investigate the visual preference appraisal.

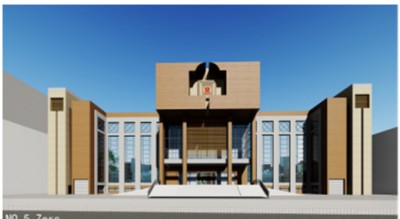 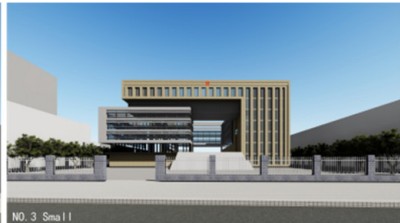 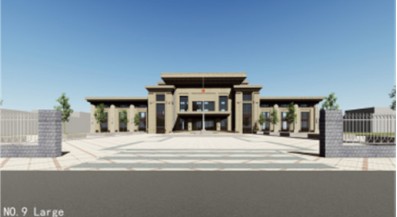

**Figure 3.** Three types of open ground in front of courthouse buildings (increasing in size from left to right).

The participants were firstly asked to fill in their demographic characteristics according to the questionnaire questions. In this experiment, three characteristics were included: gender, age, and education level (see Table 4). Then, they were asked to judge whether the buildings in these nine photos were appropriate to be courthouse buildings. On this basis, they would score the photos within the score range of 1 to 5, 1 denoting the lowest and 5 the highest. They could change their scores freely in this process. The questionnaire survey was conducted from May to August, 2020. In total, 716 participants were surveyed and 594 valid questionnaires were collected, with an effective rate of 82.9%. The implication of scores is shown in Table 5 and the demographic characteristics of participants are displayed in Table 6.

**Table 4.** The variables of different demographic characteristics and their corresponding scores.

| Demographic Characteristics | Variable |
|---|---|
| Gender | Female |
| | Male |
| | 18–34 years old |
| Age | 35–59 years old |
| | 60 and above |
| | Primary education or below |
| Education level | Secondary education |
| | College education |
| | Graduate education |

**Table 5.** The implication of photo scores.

| Score | Implication |
|---|---|
| 1 | Inappropriate |
| 2 | Not very appropriate |
| 3 | Medium |
| 4 | Appropriate |
| 5 | Very appropriate |

The data collected were analyzed with SPSS 22.0 to explore the influence of difference demographic characteristics on participants' visual preference appraisal in terms of architectural style, height–width ratio, window–wall ratio, and open ground in front of buildings. On this basis, multiple linear regression analysis was conducted to further study the influence. These analytical methods have been widely used in relevant studies [50].

**Table 6.** The demographic characteristics of participants.

| Demographic Characteristics | Variable | Number of Participants | Proportion of Participants |
|---|---|---|---|
| Gender | Female | 294 | 49.49% |
| | Male | 300 | 50.51% |
| Age | 18–34 years old | 160 | 26.93% |
| | 35–59 years old | 284 | 47.81% |
| | 60 and above | 150 | 25.25% |
| Education level | Primary education or below | 190 | 31.99% |
| | Secondary education | 194 | 32.66% |
| | College education | 164 | 27.61% |
| | Graduate education | 46 | 7.74% |

## 3. Results

### 3.1. The Overall Assessment of the Photos

Firstly, the inter-group reliability of the nine photos (see Figure 1) was tested. By way of SPSS22.0 [12], the reliability was calculated to be 0.794, displaying a relatively high reliability. Accordingly, it can be concluded that the questionnaire survey was reliable and the data obtained could be used for further detailed analysis. The mean score of each photo was denoted as S. Of all the photos, the highest mean score was 4.27, the lowest being 2.65 (the overall scoring range: 1–5); the mean score of all the photos was 3.45. The photo which scored the highest on average was a neoclassical building with an open ground in front, as is shown in Figure 4; the photo which scored the lowest on average was a post-modernist building with an open ground, as shown in Figure 5.

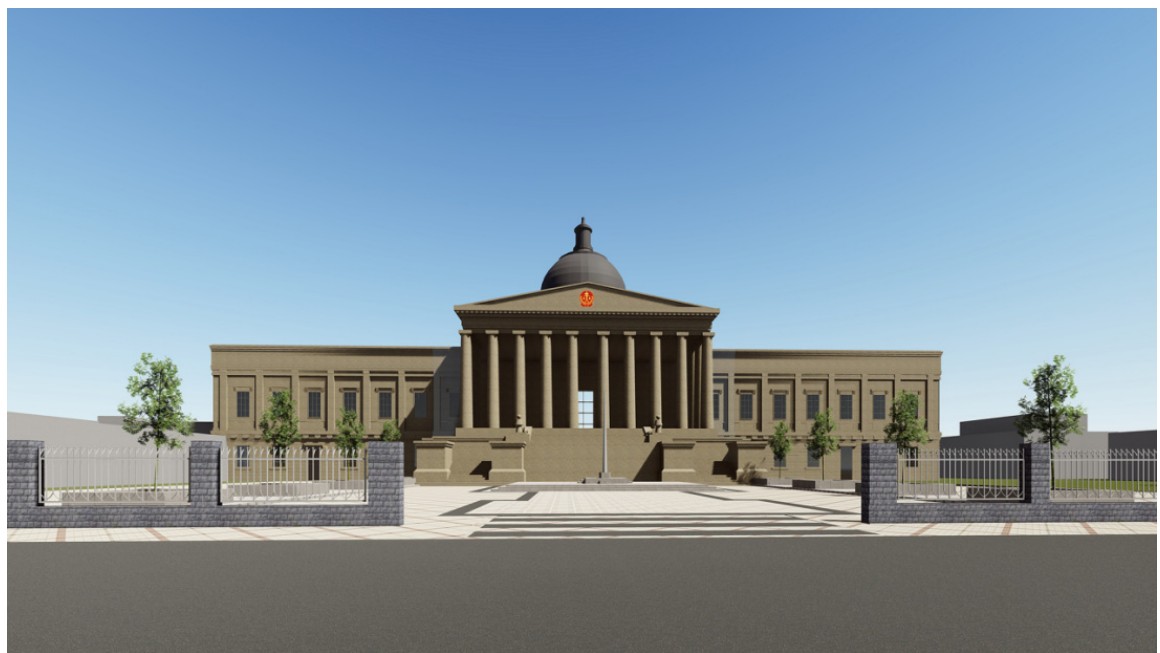

**Figure 4.** The photo with the highest mean score (No.6).

In the experiments where photos are used to replace actual landscapes, the mean score can be taken as valid data to reflect the participants' visual preference appraisal.

### 3.2. Demographic Characteristics and Visual Preference Appraisal

In order to research the correlation between demographic characteristics and visual preference evaluation, one-way analysis of variance (ANOVA) is used to test the significance of the mean difference between two or more samples. ANOVA mainly considers the mean difference between groups. The F value in ANOVA refers to the ratio of the average variance

between groups and the average variance within groups, and the P value refers to the confidence interval of the calculated test statistic F, the smaller the P value and the larger the F value, the greater the possibility to reject the null hypothesis [50]. Firstly, the relationship between demographic characteristics and visual aesthetic assessment was explored. As is revealed via the calculations, a significant difference exists in the mean scores rendered by participants with gender difference (F = 11.485, *p* = 0.01), age difference (F = 3.694, *p* = 0.025), and education level difference (F = 6.450, *p* = 0.000) as shown in Table 7.

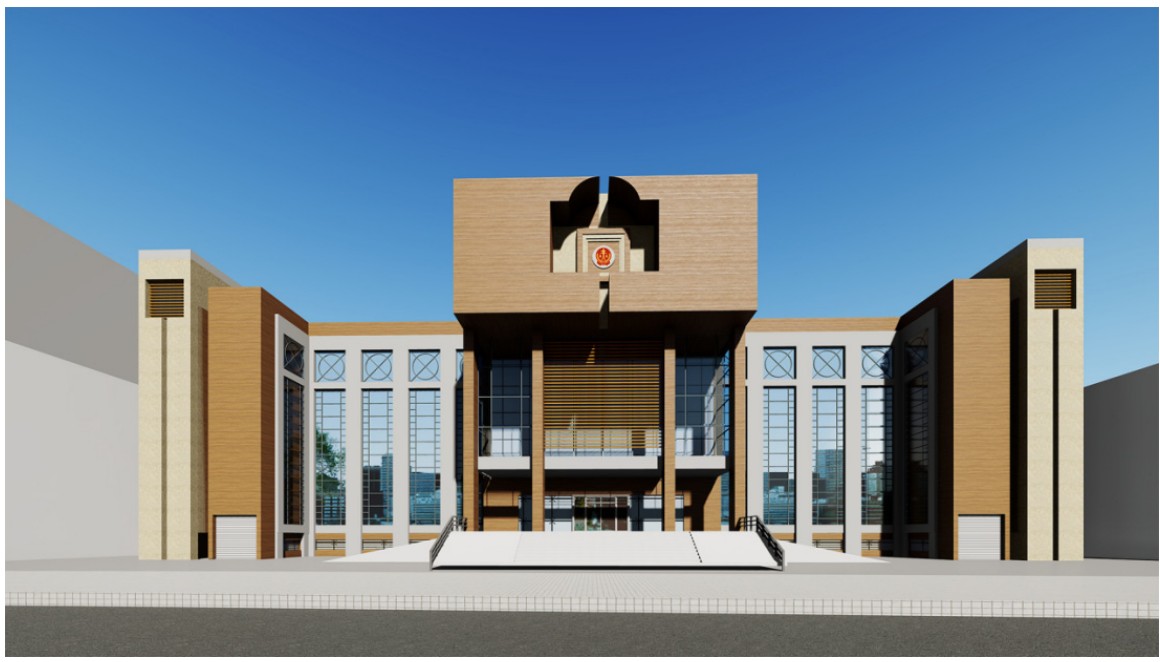

**Figure 5.** The photo with the lowest mean score (No.5).

**Table 7.** ANOVA (Between Group).

|  | **Sum of Squares** | **df** | **Mean Square** | **F** | **Sig** |
|---|---|---|---|---|---|
| Gender | 6.858 | 2 | 3.429 | 11.845 | 0.01 |
| Age | 2.814 | 2 | 1.407 | 3.694 | 0.025 |
| Education level | 6.049 | 2 | 2.417 | 6.450 | 0.000 |

Then, multiple linear stepwise regression analysis was conducted to further study the data. In the multiple regression model, gender, age, and education level were taken as independent variables and the mean score was set as dependent variable. The analysis results indicate that gender, age, and education level all exert significant influence upon the photo scores, as is shown in Table 8.

**Table 8.** Multiple linear stepwise regression analysis.

|  | **Unstandardized Coefficients** |  | **Standardized Coefficients** | **t** | **Sig.** | **Collinearity Statistics** |  |
|---|---|---|---|---|---|---|---|
|  | **B** | **Std. Error** | **Beta** |  |  | **Tolerance** | **VIF** |
| (Constant) | 3.481 | 0.03 |  | 14.389 | 0.000 |  |  |
| Age | 0.25 | 0.013 | 0.421 | 4.785 | 0.012 | 0.82 | 1.075 |
| Gender | −0.47 | 0.011 | −1.344 | −5.033 | 0.001 | 0.516 | 2.231 |
| Education level | 0.52 | 0.015 | −1.96 | −6.632 | 0.000 | 0.732 | 6.845 |

Furthermore, a collinearity analysis was conducted to the independent variables by means of the results obtained from the multiple linear regression model to study

whether reciprocal effect exists among demographic characteristics. The tolerance of age is 0.82, VIF = 1.075; that of gender is 0.516, VIF = 2.231; and that of education level is 0.732, VIF = 6.845. When VIF is over 10 or tolerance is below 0.2, collinearity exists in the model [51]. As shown in Table 8, the VIFs of independent variables are all lower than 10 and the tolerances are larger than 0.2; the residual errors are distributed normally. Therefore, it can be concluded that there is no collinearity in this model.

### 3.3. The Participants' Gender Difference and the Photos' Physical Properties

The mean scores of each photo rendered by male participants and female participants were set as dependent variables; moreover, the physical properties of photos (architectural style, height–width ratio, window–wall ratio, and open ground in front of building) were taken as independent variables. As is shown in the multiple linear stepwise regression model, the significant predictors for male and female are different (as shown in Table 9). For male participants, open ground (denoted as L), window–wall ratio (denoted as A), and height–width ratio (denoted as H) are reliable predictors. For female participants, window–wall ratio (A) and architectural style (denoted as S) are reliable predictors.

**Table 9.** The linear regression analysis of photos' physical properties for different gender groups.

| Dependent | | Unstandardized Coefficients | | Standardized Coefficients | t | Sig. | Collinearity Statistics | |
|---|---|---|---|---|---|---|---|---|
| | | B | Std. Error | Beta | | | Tolerance | VIF |
| Scores for male | Constant | 2.784 | 0.269 | | 4.125 | 0 | | |
| | L | 1.982 | 1.474 | 0.424 | 2.812 | 0.01 | 0.874 | 2.412 |
| ($R^2$ = 0.62, N = 150) | A | 0.549 | 2.315 | 0.455 | 2.944 | 0.01 | 0.784 | 2.785 |
| | H | 1.497 | 1.597 | 0.439 | 1.425 | 0.02 | 1.43 | 1.215 |
| Scores for female | Constant | 6.724 | 1.221 | | 6.638 | 0 | | |
| ($R^2$ = 0.58, N = 147) | A | 0.897 | 0.364 | 0.897 | 4.121 | 0.001 | 0.782 | 4.271 |
| | S | 1.865 | 0.562 | 0.812 | 1.365 | 0.002 | 0.415 | 5.348 |

The K-S test is named after two Soviet mathematicians, Kolmogorov and Smirnov. It is a goodness of fit test. The K-S test judges whether the observation results of the sample come from the population of the formulated distribution by analyzing the difference between the two distributions [52]. The smaller the *p* value, the smaller the probability of the original hypothesis. The coefficient of variance expansion (VIF) is a measure of the severity of complex (multiple) collinearity in multiple linear regression model. The closer the Vif value is to 1, the lighter the multicollinearity is, and vice versa. In this study, the K-S test was conducted to verify whether there is collinearity between the two models. As is shown in Table 9, the residual errors are distributed normally (male: K-S Z = 0.607, *p* = 0.601; female: K-S Z = 0.942, *p* = 0.315). Therefore, it can be concluded that there is no collinearity between the two models.

### 3.4. The Participants' Age Difference and the Photos' Physical Properties

The mean scores of each photo rendered by participants of different age groups were set as dependent variables; moreover, the physical properties of photos (architectural style, height–width ratio, window–wall ratio, and open ground in front of building) were taken as independent variables. As is shown in the multiple linear stepwise regression model, the significant predictors for participants of different age groups are different (as shown in Table 10). For participants that were 18–34 years old, height–width ratio (H), window–wall ratio, and architectural style (S) are reliable predictors. For participants that were 35–59 years old, height–width ratio (H) and architectural style (S) are reliable predictors. For participants that were over 60 years old, open ground (L), height–width ratio (H), and architectural style (S) are reliable predictors for visual preference appraisal.

A K-S test was conducted to verify whether there is collinearity among the four models. As is shown in Table 10, the residual errors are distributed normally (18–34 years old: K-S Z = 0.339, *p* = 0.124; 35–59 years old: K-S Z = 0.618, *p* = 0.271; above 60 years old: K-S

Z = 0.689, *p* = 0.481). Therefore, it can be concluded that there is no collinearity among the four models.

**Table 10.** The linear regression analysis of the photos' physical properties for different age groups.

| Model | | Unstandardized Coefficients | | Standardized Coefficients | t | Sig. | |
|---|---|---|---|---|---|---|---|
| | | B | Std. Error | Beta | | | VIF |
| 18–34 years old | (Constant) | 4.851 | 0.379 | | 4.982 | 0 | |
| | H | 0.482 | 0.792 | 0.481 | 4.192 | 0.001 | 2.874 |
| $R^2 = 0.638$ *n* = 65 | A | 0.364 | 0.835 | 0.498 | 4.325 | 0.001 | 3.524 |
| | S | 0.416 | 0.738 | 0.428 | 3.982 | 0.001 | 2.981 |
| 35–59 years old | (Constant) | 9.251 | 0.345 | | 4.356 | 0 | |
| $R^2 = 0.681$ *n* = 127 | S | 2.751 | 1.312 | 2.543 | 4.291 | 0 | 5.435 |
| | H | 1.192 | 0.942 | 1.821 | 3.982 | 0 | 4.982 |
| 60 years old or older | (Constant) | 7.312 | 0.528 | | 6.475 | 0 | |
| $R^2 = 0.714$ *n* = 55 | S | 0.478 | 1.634 | 0.639 | 4.394 | 0 | 4.361 |
| | H | 0.798 | 1.372 | 0.718 | 3.581 | 0 | 3.982 |
| | L | 1.332 | 0.782 | 1.139 | 3.612 | 0 | 4.461 |

*3.5. The Participants' Education Level Difference and the Photos' Physical Properties*

The mean scores of each photo rendered by participants of different education levels were set as dependent variables; moreover, the physical properties of photos (architectural style, height–width ratio, window–wall ratio, and open ground in front of building) were taken as independent variables. As is shown in the multiple linear stepwise regression model, the significant predictors for participants of different education levels are different (as shown in Table 11). For participants who received primary education or below, window–wall ratio (A) and architectural style (S) are reliable predictors. For participants who received secondary education, height–width ratio (H), open ground (L), and architectural style (S) are reliable predictors. For participants who received college education, open ground (L), window–wall ratio (A), and architectural style are reliable predictors for visual preference appraisal. For participants who received graduate education, open ground (L), window–wall ratio (A), and height–width ratio (H) are reliable predictors.

**Table 11.** The linear regression analysis of photos' physical properties for groups with different education levels.

| Model | | Unstandardized Coefficients | | Standardized Coefficients | t | Sig. | Collinearity Statistics | |
|---|---|---|---|---|---|---|---|---|
| | | B | Std. Error | Beta | | | Tolerance | VIF |
| Primary education or below | (Constant) | 4.125 | 0.412 | | 7.684 | 0 | | |
| $R^2 = 0.711$ *n* = 95 | S | 0.963 | 2.698 | 0.714 | 4.697 | 0 | 0.367 | 6.258 |
| | A | 0.873 | 2.183 | 0.682 | 5.291 | 0 | 0.412 | 5.821 |
| Secondary education | (Constant) | 3.856 | 0.392 | | 6.854 | 0 | | |
| $R^2 = 0.694$ *n* = 97 | H | 0.426 | 1.482 | 0.347 | 2.528 | 0.02 | 0.521 | 4.528 |
| | L | 0.652 | 0.831 | 0.472 | 2.481 | 0.01 | 0.471 | 3.752 |
| | S | 0.572 | 1.382 | 0.892 | 3.412 | 0 | 0.772 | 4.582 |
| College Education | (Constant) | 4.528 | 0.582 | | 2.257 | 0 | | |
| $R^2 = 0.639$ *n* = 82 | L | 0.471 | 1.052 | 0.428 | 4.747 | 0 | 0.782 | 3.127 |
| | A | 0.398 | 2.821 | 0.376 | 3.471 | 0 | 0.672 | 2.852 |
| | S | 0.576 | 0.582 | 0.641 | 4.716 | 0 | 0.418 | 3.251 |
| Graduate education | (Constant) | 6.483 | 0.439 | | 3.598 | 0 | | |
| $R^2 = 0.639$ *n* = 23 | L | 0.386 | 2.551 | 0.536 | 4.819 | 0 | 0.834 | 3.982 |
| | H | 0.471 | 3.412 | 0.672 | 4.112 | 0 | 0.749 | 3.192 |
| | A | 0.376 | 3.598 | 0.441 | 2.421 | 0 | 0.582 | 2.932 |

A K-S test was conducted to verify whether there is collinearity among the four models. As is shown in Table 11, the residual errors are distributed normally (primary education and below: K-S Z = 0.678, *p* = 0.142; secondary education: K-S Z = 0.571, *p* = 0.352; college education: K-S Z = 0.648, *p* = 0.431; graduate education: K-S Z = 0.572, *p* = 0.339). Therefore, it can be concluded that there is no collinearity among the four models.

From the linear regression analysis, it can be seen that no definite conclusion has been achieved about the visual preference difference caused by demographic characteristics in

terms of architecture or environment. One important reason for this is that there exists a reciprocal effect among the demographic characteristics. However, the reciprocal effect is hard to define clearly in that the participants have different growth environments, cultural backgrounds, and life experiences. Consequently, to test whether there exists a reciprocal effect, a regression model has to be used. By means of multiple linear regression models, this experiment reveals that there is no reciprocal effect among gender, age, and education level.

## 4. Discussion

### 4.1. Demographic Characteristics and Visual Preference Appraisal

People with different demographic characteristics may assess the same group of photos differently. Participants in different age groups might rate the same building image differently [32,53]. As the age of participants increases, the mean score they render upon the photos declines slightly [33]. This is in line with the results of this research. As shown in Table 8, the participants that were over 60 years old scored the photos the lowest. The reason for this may be that the senior group attach more importance to the solemnity of courthouse buildings when they assess the photos [54]. They held that neoclassical buildings were more suitable to serve as courthouse architectures for its grand size, elegance, and rhythm; moreover, out of a strong sense of nostalgia, they would compare the photos with their living and working environment and then render relatively higher assessment upon the neoclassic courthouse buildings. As for the younger participants, they witnessed the rapid development of Chinese urbanization and, thus, tended to be more tolerant to multiple architectural forms [55]. Accordingly, compared with older participants, they scored higher on the visual preference appraisal of the same group of photos. Different age groups may produce different visual preference appraisals due to the multiple influence of living environment and life experience [32,56]. This is also consistent with the results of this research. Therefore, when judging the visual preference appraisal of different age groups, we need to take the difference in their culture background and living environment into consideration rather than simply draw the conclusion.

Gender difference also leads to different visual preference appraisals [57–59]. Females scored the external form of architectures lower than males do [59]. This is inconsistent with the results of this research. The reason for this difference may lie in the fact that as a judicial symbol, courthouse buildings appear to be the same in the eye of male and female for its solemnity and authority. As a symbol of power, judicial architecture will enhance people's sense of class and weaken their gender concept [60]. In this sense, it is highly reasonable that there is no significant difference between males and females in their visual preference appraisals.

Citizens with different educational backgrounds could probably show different preference in the elements of building or landscapes [23,61]. Previous studies also maintained that education level would influence people's visual preference appraisal of landscapes; participants with a lower education level scored the neoclassical courthouse buildings higher [34]. This is in line with the results of this research. Participants with a lower education level are more familiar with courthouse buildings of neoclassical style; on the other hand, they are relatively unfamiliar with other styles. In this case, they would render relatively lower scores for courthouse buildings of other styles. By contrast, participants with higher education levels have already received certain knowledge of visual aesthetics in their learning process [62]; moreover, they have seen or visited many buildings and landscapes of various styles. Therefore, when they assess the external form of Chinese courthouse buildings, they would be more comprehensive and, thus, render higher scores for non-neoclassical buildings.

All in all, although no definite final conclusion has ever been drawn on the influence of different demographic characteristics on visual preference appraisals, this research maintained that research objects, characteristics of the times, and education level should be the main consideration.

*4.2. Demographic Characteristics and Physical Characteristics of Photos*

4.2.1. Age and Physical Characteristics of Photos

Some research revealed that different age groups would produce different visual preference appraisals of the external form of buildings [63]. This is consistent with the experimental results of this research. As has been revealed in this research, participants that were 18–34 years old gave their highest scores to the courthouse buildings with a large window–wall ratio and small height–width ratio. The reason for this result may be that this age group is familiar with Chinese traditional courthouse buildings and that the increase of window area may bring forth some fresh feelings [64]. Participants that were 35–59 years old preferred buildings of neoclassical style with a smaller height–width ratio as the courthouse buildings. This middle-aged group of participants attached more importance to the symbolic meaning and functionality of courthouse buildings. Participants that were over 60 years old held that buildings of neoclassical style with smaller height–width ratio and large open ground in front were more suitable to serve as courthouse architecture. The senior participants also prioritized the symbolic meaning and functionality of courthouse buildings. The neoclassical style features sculptural forms, elegance, solemnity, harmony, and a clear relationship between the principal and the subordinate [65]. Mostly, its symmetric design can better reflect the solemnity and justice of judiciary. Neoclassical architectures usually adopt the classic order composition of the ancient Greece and Roman period, which can better display the majesty and solemnity of buildings [66].

4.2.2. Gender and Physical Characteristics of the Photos

Gender difference also leads to different visual preference appraisals [23]. This is consistent with the results of this research. As is revealed through the experiment, females score the Chinese courthouse buildings with a small height–width ratio higher in that they think buildings with gentle forms are more suitable to serve as courthouse buildings; similarly, they think that if the courthouse building is relatively low, it could be friendlier. By contrast, males prefer buildings with a large window–wall ratio and a small height–width ratio. In terms of open ground in front of buildings, both male and female participants maintained that buildings with large open ground were more suitable as courthouse buildings.

4.2.3. Education Level and Physical Characteristics of Photos

People of different education levels would produce different visual preference appraisals of the external forms of buildings [67]. For participants who receive secondary education or below, their impression on the courthouse mostly comes from their elders and local courthouse buildings; in addition, they have quite rare opportunities to contact buildings of other styles in their daily life. Accordingly, they would prefer neoclassic buildings when assessing the photos. On the other hand, the Chinese courthouse buildings of neoclassical style appear to be solemn and majestic, which satisfies the imagination of these participants upon courthouse buildings. By contrast, participants with college or graduate education showed no remarkable preference for the architectural style of courthouse buildings. The reason is that they possess a more comprehensive knowledge of laws and have more contacts with courthouse buildings in their daily life, thus lacking a feeling of awe to courthouses [68,69]. Participants with secondary education or below maintained that buildings with a large window–wall ratio were more suitable to serve as courthouse buildings. Comparatively, they showed no significant preference for the height–width ratio of Chinese courthouse buildings. For them, a large window–wall ratio would make the courthouse livelier and less oppressing. On the contrary, participants with college or graduate education preferred buildings with small height–width ratio to serve as a courthouse. The reason for this may lie in the fact that participants with higher education levels would be more rational when assessing the physical properties of the photos [70]. They would choose some physical properties easy to be caught visually and consider the issue from the perspectives of energy-saving performance, functionality, and practicability. All the

participants who received a secondary education and above maintained that buildings with a large open ground in front would be more suitable for courthouse buildings.

**5. Conclusions**

Courthouse architecture is in essence one kind of judicial building. Its design cannot merely rely on a sudden artistic inspiration. In other words, its design must take its judicial function into consideration. The courthouse architecture should have a distinctive style, which can help people to distinguish it from the surrounding buildings [71]. At present, China is in a transition period of economic structure. Accordingly, the courthouse buildings in different regions and cities display a drastic variation in terms of architecture style. This research focused on the relationship between the physical properties of Chinese courthouse buildings and visual preference appraisal. As is revealed in this research, architectural style, height–width ratio, window–wall ratio, and open ground in front of the building all exert certain influence on people's visual preference appraisal; moreover, the neoclassical courthouse building with a large window–wall ratio, small height–width ratio, and large open ground is the most favorite among the participants.

This research is can help architects to better understand Chinese people's views about courthouse buildings and grasp what physical properties are more important to participants of different demographic characteristics evaluating courthouse buildings. Although this research did not conduct an in-depth analysis of whether there exists any correlation among participant groups of different gender, age, and education level, the experimental and analytical results obtained could offer architects and designers valuable and practical reference in their actual practices; to some degree, these results can also indicate the changing trend of Chinese people's visual preference appraisal of courthouse buildings.

This research has some limitations in terms of experimental methodology and treatment of environmental factors in courthouse buildings. In future research, we will consider the influence of the building's surrounding environment, physical environment, and the impact of special groups on the visual preference evaluation of Chinese courthouse buildings, and use eye-tracking or facial reaction diagnosis to extend research to real architecture and space at different times of the year.

**Author Contributions:** Conceptualization, J.P.; methodology, X.W.; software, Y.Y.; validation, J.P. and Y.Y.; formal analysis, Y.Y.; investigation, Y.Y.; resources, X.W.; data curation, Y.Y.; writing—original draft preparation, J.P.; writing—review and editing, J.P.; visualization, X.W.; supervision, J.P.; project administration, C.H.; funding acquisition, C.H. All authors have read and agreed to the published version of the manuscript.

**Funding:** This research received no external funding.

**Institutional Review Board Statement:** Not applicable.

**Informed Consent Statement:** Informed consent was obtained from all subjects involved in the study.

**Data Availability Statement:** Not applicable.

**Conflicts of Interest:** The authors declare no conflict of interest. The funders had no role in the design of the study; in the collection, analyses, or interpretation of data; in the writing of the manuscript; or in the decision to publish the results. This paper represents the opinions of the author(s).

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
