# Peer review of "Research on Visual Preference of Chinese Courthouse Architecture Appearance"

_buildings, doi:10.3390/buildings12050557_

Round 1
Reviewer 1 Report
The manuscript analyses public preferences on the visual appearance of courthouse buildings in China. The title of the paper clearly describes the content. The methodology followed is sound and the discussion of results is based on a statistical analysis of the data. Authors must review the reference list and clarify several issues regarding the development of the research.
Particular remarks are quoted below:
- Abstract: The sentence “As is indicated,….” This sentence repeats ideas previously stated. I suggest replacing it with another one summarizing the main conclusions of the article.
- A good number of the references cited in the manuscript are not correctly included in the list of references (i.e. Robert Jacobs, 1994; Resnik et al, 2014; Yuan & Youbing, 2017, etc.). Authors must review the complete list of references. Please, adapt the format of the citations to that indicated in the ‘Instructions for authors’.
- Pp. 2. “… four characteristics are considered, namely, age, gender, education level and place of residence.” . I could not find in your research an analysis of the influence of the place of residence and people´s visual preference for courthouse buildings.
- Pp. 3. “Some scholar used AR method”. Please, explain the meaning of this abbreviation.
- Architectural style of courthouse buildings: The main architectural features of each style considered in the article (modernist, neoclassical and post-modernist) should be properly described. I do not understand the reason why no Chinese-style courthouse building was included in the study. Please justify it adequately.
- Please describe in some more details the procedure followed to produce the models in SketchUp.
- Figure 1: Please, number the pictures properly and indicate the architectural style represented by each photo.
- Table 2: True dimensions of the buildings should be better expressed in meters.
- “To calculate the size of open ground in front of the courthouse building”: In my opinion, a representative example of each category would be of interest.
- Table 6: The total sum of participants by age is 608 (and not 594). Please, check it.
- Some additional information should be included for a better understanding and interpretation of the results obtained in the statistical analysis. So, for example, the level of confidence for the different tests or the critical F values in ANOVA should be stated. On the other hand, the fundamentals of the Kolmogorov–Smirnov (K–S) test should be shortly described, as well as the way to interpret the different statistical indicators.
- Pp. 10. (… Under 17 years old: ) Was this age group also included in the analysis? Please, explain it.
- Additional research would be of interest to explain the reasons of the different visual preferences between groups.
Author Response
1.Abstract: The sentence “As is indicated,….” This sentence repeats ideas previously stated. I suggest replacing it with another one summarizing the main conclusions of the article.
Answer: We have deleted the duplicate part in this section and described the conclusion more specifically, add “the neoclassical courthouse building with large window-wall ratio, small height-width ratio and large open ground is the most favorite among the participants” to the abstract, and the correction has been made in the manuscript.
2.A good number of the references cited in the manuscript are not correctly included in the list of references (i.e. Robert Jacobs, 1994; Resnik et al, 2014; Yuan & Youbing, 2017, etc.). Authors must review the complete list of references. Please, adapt the format of the citations to that indicated in the ‘Instructions for authors’.
Answer: We reviewed the draft again according to the logical requirements of the paper,the references have been correctly included in the manuscript reference list according to the format and review comments specified in the "instructions to authors".
3.Pp. 2. “… four characteristics are considered, namely, age, gender, education level and place of residence.” . I could not find in your research an analysis of the influence of the place of residence and people´s visual preference for courthouse buildings.
Answer: We are very sorry that this was a mistake in our writing. Considering that the calculation of the paper is too complex after adding the factor of place of residence, we did not take the factor of “place of residence” into account in the follow-up research, but the previous article was not changed in time. And the correction has been made in the manuscript.
4.Pp. 3. “Some scholar used AR method”. Please, explain the meaning of this abbreviation.
Answer: In this method, a number of 3D virtual rectangular objects with a scale are located on the grid of 3D geographical model. And then, the CG models are displayed in an overlapping manner with the actual landscape from multiple viewpoints using the AR technology. The user measures the maximum invisible height for each rectangular object at a grid point. The correction has been made in the manuscript.
5.Architectural style of courthouse buildings: The main architectural features of each style considered in the article (modernist, neoclassical and post-modernist) should be properly described. I do not understand the reason why no Chinese-style courthouse building was included in the study. Please justify it adequately.
Answer: In the course of our visit, investigation and online information inquiry, Chinese style courthouse building was not found in today's China's primary court, intermediate court and high court. Although some courthouse buildings have some decorative characteristics of Chinese architecture, the main structure of the building is still a modern architectural structure system, so we still think that it belongs to modern buildings. Actually ,after experiencing China's reform and opening up in the 1980s, the freedom of speech was open. The style of European and American legal system was widely introduced into China's courthouse building, the concept of the rule of law, judicial institutions and judicial uniforms, and has been used to this day.
6.Please describe in some more details the procedure followed to produce the models in SketchUp.
Answer: We have added the details and steps of SketchUp modeling in the manuscript according to the review opinions, including selecting the perspective of the main facade of the building for modeling, retaining the main part of the building and the enclosure structure, and explaining the rendering process.
7.Figure 1: Please, number the pictures properly and indicate the architectural style represented by each photo.
Answer: We have added picture numbers to the manuscript drawings and described the style represented by each picture. And use "No." in the later figures and tables to make the context more closely connected, this modification makes the image information expression more intuitive.
8.Table 2: True dimensions of the buildings should be better expressed in meters.
Answer: Your comments are very effective for the manuscript, this makes the paper data more scientific. We revised Table 2 using court real building dimensions in meters and proofread the relevant locations involving the item data in the manuscript.
9.“To calculate the size of open ground in front of the courthouse building”: In my opinion, a representative example of each category would be of interest.
Answer: Your comments are constructive, it makes the manuscript description more vivid. We have added ”Figure. 3 three types of open ground in front of courthouse buildings” to the manuscript,three types of open ground in front of courthouse buildings are made more intuitive through the image.
10.Table 6: The total sum of participants by age is 608 (and not 594). Please, check it.
Answer: We are very sorry, it was a mistake in our writing. After proofreading, we found that the "35-59 years old" column is 284, not 298.The total sum of participants by age is still 594, and the correction has been made in the manuscript.
11.Some additional information should be included for a better understanding and interpretation of the results obtained in the statistical analysis. So, for example, the level of confidence for the different tests or the critical F values in ANOVA should be stated. On the other hand, the fundamentals of the Kolmogorov–Smirnov (K–S) test should be shortly described, as well as the way to interpret the different statistical indicators.
Answer: Your comments are helpful to our manuscript, to better explain the different metrics, we added two descriptions about the "Kolmogorov–Smirnov (K–S) test" and "ANOVA", briefly introducing their definitions and rationale, enabling readers to better understand the process of our experiments and the meaning expressed by the experimental results. The correction has been made in the manuscript.
12.Pp. 10. (… Under 17 years old: ) Was this age group also included in the analysis? Please, explain it.
Answer: We are very sorry that this was a mistake in our writing. We did not include a group under 17 years old in our study. We think that most of the people under 17 years old are students, who have been in the campus environment for a long time, for the court concept and function is not very familiar with, for the courthouse building lacks basic cognitive ability. We have removed the unrelated description sections of the manuscript.
13.Additional research would be of interest to explain the reasons of the different visual preferences between groups.
Answer: Thank you for your opinions and suggestions for our article. The additional research you proposed is of great significance to the different visual preferences between groups. In future research, we will take it as the key research direction.
Reviewer 2 Report
I have reviewed this manuscript treating the visual preferences of courthouses in China
The authors discuss an interesting problem in the sphere of visual preferences. However, the research is made with photographs and not with the real buildings, it could be necessary to augment the degree of perception of the places by interviewing passers-by. Also the opinion of the users of these facilities could be determinant.
The paper describes the background in a concise and satisfactory manner but not sufficiently architectural. Data on the locations of the buildings and its weather are entirely missing. It is known that the perception of a building changes greatly in different sky and weather conditions.
The statistical analysis on the photographs is correct from a sociological point of view. One wonders if more insights could be gained from a 3D analysis of the buildings and courthouses complexes instead of just 9 frontal 2D pictures shown in the analysis.
The results section is slightly insufficient in this sense and does not lead to clear conclusion. The concept of EPP should be better defined.
The conclusions of the manuscript should be enhanced with the intentions to continue the research and extend it to the real buildings and the spaces under consideration in different times of the year. A suggestion for new designs of the experiments is advised.
Summary of evaluation: This article is interesting and deals with useful matter for research on design and architectural composition. However, I find relevant faults in the study and I recommend major revisions are effected to the article in the way explained previously.
Author Response
- The paper describes the background in a concise and satisfactory manner but not sufficiently architectural. Data on the locations of the buildings and its weather are entirely missing. It is known that the perception of a building changes greatly in different sky and weather conditions.
Answer:The question you have considered are of great reference value,from the perspective of design, it is often difficult for architects to respond to different weather conditions in the design scheme, and introducing different weather conditions into the experiment will also make the experiment too complex to get accurate and effective conclusions. So in order to provide valuable references to the architect, these factors were also excluded in our experiments. In the future research, we will conduct a more thorough research and thinking about the weather factors.
- The statistical analysis on the photographs is correct from a sociological point of view. One wonders if more insights could be gained from a 3D analysis of the buildings and courthouses complexes instead of just 9 frontal 2D pictures shown in the analysis.
Answer:This is a great suggestion, although the replacement of physical building with pictures is a common experimental method, but we also realize that this experimental method really has great limitations, in the following experiments we will try our best to enrich our experimental method and experimental equipment.
- The results section is slightly insufficient in this sense and does not lead to clear conclusion. The concept of EPP should be better defined.
Answer:Your proposal is very constructive.We added representations of EPP to the manuscript and added relevant explanations about Visual impact and Visual evaluation method.
- The conclusions of the manuscript should be enhanced with the intentions to continue the research and extend it to the real buildings and the spaces under consideration in different times of the year. A suggestion for new designs of the experiments is advised.
Answer:Following your proposal, we strengthen the conclusion section of the manuscript and make recommendations for a new design, and your idea of extending research to real architecture and space at different times of the year is significant.In the future research, we will focus on it as a key research direction.
- Summary of evaluation: This article is interesting and deals with useful matter for research on design and architectural composition. However, I find relevant faults in the study and I recommend major revisions are effected to the article in the way explained previously.
Answer:Thank you for your opinions and suggestions for our article.We revised the manuscript according to the draft opinions, and supplemented the corresponding explanations.Your valuable comments have important reference value for our paper revision and further research in the future.
Reviewer 3 Report
The article deals with a socially interesting topic. Many famous authors and politicians have written about style preferences in the context of architecture. The theme of style preferences has also become the basis for the well-known novel by Ayn Rand "The Fountainhead", which shows the importance of this issue. There was no insight into the topic's historical context in the introduction. I also propose to mention the reflections of the philosopher Paul-Michel Foucault's "Discipline and Punish".
The beginning, referring to the opinion of the former US President, gives the whole dynamism and puts the reader in a good mood. Do Chinese politicians and architects comment on this type of style preference?
The main objection is the lack of ensuring the consistency of the methodology - there is no indication of what cities (size/number of inhabitants were the subject of the search), please add a reconnaissance map.
-Do the analyzed buildings come from the same period? A table with basic data and grouped photos of the buildings that were used to create the classification should be placed. This should be in the open repository to be able to evaluate the selection!
-Looking at the research group. Were the participants asked whether they were architects, town planners, judges (...) whether they live or walk in front of the court every day (...). Do the inhabitants live in a given place for a long time, were they born - in the countryside or the city? Have they ever had a conflict with the law? Have volunteers testified in court once and for how long? These are the elements that can influence your judgment.
-Have participants with potential visual impairments been excluded from the study?
-In how many places/cities were sets of photos shown (please mark them on the map, and if there was one place, please indicate it specifically?) If there was a court building in a certain style in the city, was it included in the set and was it assessed in the same as the others?
-I don't understand table 4 - why are points awarded for being a woman, a man, etc.
-Why are the research groups divided into 18-34 / 35-59 / 60+?
-In section 1.3 it is stated that the environment influences the perception of a building. Visualizations differ in terms of their surroundings - e.g. a fence. The frame presented to people should be guided by some logic (e.g. it always occupies the central part and has a width of the image). Buildings cannot have any other lighting, sometimes to the right, then to the left, under a sharp and gentle angle. Some of the visualizations have a darker sky, some of the visualizations are green and others do not, the colors of the facade are different, the distance of the observer from the buildings is different ... If we compare the influence of the window surface on the preferences, maybe the initial building surface shown on the printout should be the same? These are fundamental flaws in the methodology. There are so many variables that we de-facto do not know what made the choice.
-Many aspects discussed in the discussion are not supported by research results or external literature support. “In this sense, it is highly reasonable that there is no significant difference between males and females in their visual preference” there are many eye-tracking research about that…
-Salingaros writes about visual preferences (see approaching Imotion webinar). Rusnak researches the culture and perception of architecture, see publications and youtube.
-Buildings located in culturally and economically diverse areas of the country make it difficult to provide one aesthetic solution.
-Interesting statistical analyzes were performed. It is described very carefully, but, in my opinion, the data was inconsistently collected.
The preference studies are interesting, if published (!?) they should be discussed carefully at the end in terms of the disadvantages noticed and necessarily repeated.
In the next study, I propose to supplement the search with eye-tracking or facial reaction diagnosis.
Author Response
- The article deals with a socially interesting topic. Many famous authors and politicians have written about style preferences in the context of architecture. The theme of style preferences has also become the basis for the well-known novel by Ayn Rand "The Fountainhead", which shows the importance of this issue. There was no insight into the topic's historical context in the introduction. I also propose to mention the reflections of the philosopher Paul-Michel Foucault's "Discipline and Punish".
Answer:The two works you mentioned are very much in line with the background of our paper. We have added the relevant discussion of these two works to the preface of the manuscript, making the background introduction more rich and comprehensive.
- The beginning, referring to the opinion of the former US President, gives the whole dynamism and puts the reader in a good mood. Do Chinese politicians and architects comment on this type of style preference?
Answer:We have done relevant research at the beginning of writing the paper. Actually, few relevant policies or architects in China have put forward clear and clear views on the design of court buildings since the beginning of this century, which is why we planned this experiment.
- The main objection is the lack of ensuring the consistency of the methodology - there is no indication of what cities (size/number of inhabitants were the subject of the search), please add a reconnaissance map.
Answer:Your view is very correct, in fact, we also thought of this problem when writing the paper, the specific position on the map is very clear and intuitive, but considering the size of the map will take up too much space, we hope to solve the problem by marking the specific position on the bottom of the “Figure 1”.
- Do the analyzed buildings come from the same period? A table with basic data and grouped photos of the buildings that were used to create the classification should be placed. This should be in the open repository to be able to evaluate the selection!
Answer:Your comments are very important. We summarized the architectural style, window wall ratio, height width ratio, size of open ground, building prototype address and operation time of the nine pictures in "Figure 1",and each drawing is numbered.
- Looking at the research group. Were the participants asked whether they were architects, town planners, judges (...) whether they live or walk in front of the court every day (...). Do the inhabitants live in a given place for a long time, were they born - in the countryside or the city? Have they ever had a conflict with the law? Have volunteers testified in court once and for how long? These are the elements that can influence your judgment.
Answer:Your proposal is constructive, in fact, there are a large number of demographic factors may affect the respondents on the evaluation of the court, but the physical elements and demographic characteristics of courthouse building on the evaluation of courthouse building, in order to ensure the clarity of the experiment, only gender, age and education level were selected as research subjects, we will study more demographic factors in the following study.
- Have participants with potential visual impairments been excluded from the study?
Answer: We are sorry that we did not consider this in the experiment, but actually no people with visual disabilities participated in our experiment.In future studies, we will add this factor to the research to make our paper more comprehensive.
- In how many places/cities were sets of photos shown (please mark them on the map, and if there was one place, please indicate it specifically?) If there was a court building in a certain style in the city, was it included in the set and was it assessed in the same as the others?
Answer:Your suggestion is very important. In order to avoid the special feelings that urban residents may have about the courthouse buildings in their city, we chose Xuzhou, Nanjing and Zhengzhou to conduct a questionnaire survey.And the correction has been made in the manuscript.
- I don't understand table 4 - why are points awarded for being a woman, a man, etc.
Answer:What we actually wanted to express was that for statistical analysis we assigned values for men and women, only to facilitate calculation in the software and did not affect the final outcome analysis.
- Why are the research groups divided into 18-34 / 35-59 / 60+?
Answer:This is the classification we conducted following previous studies, such as articles“Demographic groups” ,”Differences in visual preference for vegetated landscapes in urban green space ” ,“The influence of newly built high-rise buildings on visual impact assessment of historic urban landscapes: a case study of Xi’an Bell Tower”The ages of the respondents were similarly classified, and in fact, in the social context of China, choosing 18-34 / 35-59 / 60 + is also a relatively representative age segment.
- In section 1.3 it is stated that the environment influences the perception of a building. Visualizations differ in terms of their surroundings - e.g. a fence. The frame presented to people should be guided by some logic (e.g. it always occupies the central part and has a width of the image). Buildings cannot have any other lighting, sometimes to the right, then to the left, under a sharp and gentle angle. Some of the visualizations have a darker sky, some of the visualizations are green and others do not, the colors of the facade are different, the distance of the observer from the buildings is different ... If we compare the influence of the window surface on the preferences, maybe the initial building surface shown on the printout should be the same? These are fundamental flaws in the methodology. There are so many variables that we de-facto do not know what made the choice.
Answer: Your advice is very important and correct. We try to use some methods to reduce Angle, light, background and the influence of visual preference evaluation, such as we did not use the original photos, but with PS processing and unified rendering method of building background, and then have a unified effect, and we take the building facade pictures as visual preference evaluation material, to avoid the influence of different angles on visual preference evaluation, these methods may not be very perfect. In the future research, we will consider the influence of the building's surrounding environment, physical environment and the impact of special groups on the visual preference evaluation of Chinese courthouse buildings, and use eye-tracking or facial reaction diagnosis to extend research to real architecture and space at different times of the year.
- Many aspects discussed in the discussion are not supported by research results or external literature support. “In this sense, it is highly reasonable that there is no significant difference between males and females in their visual preference” there are many eye-tracking research about that…
Answer:Your opinion has a very important reference value, we added external literature support in the discussion section to improve the persuasion of the paper, and added more argument support in the section "Gender difference".
- Salingaros writes about visual preferences (see approaching Imotion webinar). Rusnak researches the culture and perception of architecture, see publications and youtube.
Answer:Your suggestion is very constructive, we find and study Salingaros and Rusnak 's discussion on"visual preferences" and "culture and perception of architecture" .And The relevant summary has been quoted in the”1.2 Visual preference appraisal” of the manuscript.
- Buildings located in culturally and economically diverse areas of the country make it difficult to provide one aesthetic solution.
Answer:It is indeed a difficult thing to propose a solution that can provide extensive guidance on architectural form design, and the purpose of our experiment is to discuss the influence of some court architectural elements on the evaluation of their visual preferences, so as to provide reference for architects when carrying out relevant design.
- Interesting statistical analyzes were performed. It is described very carefully, but, in my opinion, the data was inconsistently collected.
Answer:We are very sorry, it was a mistake in our writing. After proofreading, We found problems with data collation in Table 6, the "35-59 years old" column is 284, not 298.The total sum of participants by age is still 594,and the correction has been made in the manuscript.
- The preference studies are interesting, if published (!?) they should be discussed carefully at the end in terms of the disadvantages noticed and necessarily repeated.In the next study, I propose to supplement the search with eye-tracking or facial reaction diagnosis.
Answer:Your point is correct,Our experimental method does have limitations (including not considering the surroundings of the building, the impact of the physical environment on the building, and the preference of special people for the buildings), and we modified the conclusion section to clarify these limitations. Thank you for your opinion, and we will further enrich our research methods and experimental facilities in the following study.
Round 2
Reviewer 1 Report
I thank the authors for their corrections. Several minor remarks are included in the attached PDF.

Author Response
Answer:Thank you very much for your kind notes in the manuscript. Your modification opinions are rally helpful for the manuscript. We have carefully revised that according to your suggestion, supplemented the content of the article, and adjusted the format of the article. And also, the correction has been made in the manuscript.
Reviewer 2 Report
I have reviewed this manuscript treating the visual preferences of courthouses in China.
The references presented by the authors are interesting. They should also consider the Panopticon Writings of J. Bentham to complement their introduction.
The authors discuss an interesting problem in the sphere of visual preferences. Although the research and methodology was done through photographs the authors have improved significantly the scientific level of their paper.
The manuscript describes the background in a concise and satisfactory manner but not necessarily architectural with which the conclusions are somehow limited. Data on the locations of the buildings have been precisely incorporated. It is known that the perception of a building changes greatly in different sky and weather conditions and the authors have recognized this fact but left it out for future researchers. I just have to say that in my experience of climate in these areas, the sky is rarely perceived as bright blue as presented in the beautiful reconstructions crafted by authors.
The statistical analysis on the photographs is correct from a sociological point of view. The authors have augmented the insights of their experiments, some minor parts remain however doubtful.
The results section leads to appropriate conclusions which have been enhanced with the intentions to continue the research and extend it to the real buildings and the spaces under consideration in different times of the year, though this has not been completely performed by the authors.
Summary of evaluation: This article is interesting and deals with useful matter for research on design and architectural composition, the additions performed are sufficient and I recommend acceptance after minor reviews in the sense described in the above paragraphs.
Author Response
I have reviewed this manuscript treating the visual preferences of courthouses in China.The references presented by the authors are interesting. They should also consider the Panopticon Writings of J. Bentham to complement their introduction.
Answer: Your suggestion is very helpful. This enriches our introduction a lot. We carefully searched the relevant works of J. Bentham and cited them in the corresponding part of the manuscript.
The manuscript describes the background in a concise and satisfactory manner but not necessarily architectural with which the conclusions are somehow limited. Data on the locations of the buildings have been precisely incorporated. It is known that the perception of a building changes greatly in different sky and weather conditions and the authors have recognized this fact but left it out for future researchers. I just have to say that in my experience of climate in these areas, the sky is rarely perceived as bright blue as presented in the beautiful reconstructions crafted by authors.
Answer: The problem you mentioned is very important. We will choose more neutral colors for the architectural background in the follow-up research to highlight the building itself and further reduce the influence of other interference factors.
The statistical analysis on the photographs is correct from a sociological point of view. The authors have augmented the insights of their experiments, some minor parts remain however doubtful.
Answer: Your comments are very helpful. We have further revised and supplemented the corresponding parts in the manuscript, and explained the meaning of the indicators in the experimental process. I hope the final conclusions can be convincing.
The results section leads to appropriate conclusions which have been enhanced with the intentions to continue the research and extend it to the real buildings and the spaces under consideration in different times of the year, though this has not been completely performed by the authors.Summary of evaluation: This article is interesting and deals with useful matter for research on design and architectural composition, the additions performed are sufficient and I recommend acceptance after minor reviews in the sense described in the above paragraphs.
Answer: Thank you very much for your opinions and suggestion. These suggestions perfect and enrich our manuscript. In the future research process, we will focus on referring to your opinions for more in-depth exploration and Research on architectural environment and experimental methods.
Reviewer 3 Report
The authors devoted a lot of time to proofreading the text. However, the correction cannot remove the underlying methodological errors. This issue is mitigated by the self-criticism of the method used. If the editor is planning to allow the publication of defects, they should be extracted as a separate and important point.
The work of Salingaros has the same disadvantage as the work of the authors. Stimuli are not homogeneous. Even out of cleer curiosity, please look for research in Boston.
The cited study by Rabiega and Rusnak mainly points to potential use in education. Other publications more fully indicate the potential of public consultations and the study of their preferences.
Thank you for your extremely kind and patient answers and changes. After expansion, you should take care of the fluidity of the narrative. The text is long, so it is very important to keep reader's attention
Correction of typos, missing or extra spaces and empty fragments are required.
I hope that the next tests - according to the declarations will be perfect.
Author Response
The authors devoted a lot of time to proofreading the text. However, the correction cannot remove the underlying methodological errors. This issue is mitigated by the self-criticism of the method used. If the editor is planning to allow the publication of defects, they should be extracted as a separate and important point.
Answer: Your opinion is very correct. Our experimental method does have limitations. Therefore, in the final conclusion, we emphasize the shortcomings of the experimental method and the treatment of environmental factors, and will take this aspect as the key content for in-depth research in future research.
The work of Salingaros has the same disadvantage as the work of the authors. Stimuli are not homogeneous. Even out of cleer curiosity, please look for research in Boston.
Answer: This problem also confuse us a lot, and we have replaced the literature according to your proposal and quoted the work of Sussman A and Ward J to improve the pertinence of the manuscript.
The cited study by Rabiega and Rusnak mainly points to potential use in education. Other publications more fully indicate the potential of public consultations and the study of their preferences.
Answer: Your suggestion is very helpful. According to your suggestion, we added visual preference appraisal literature on public consultation to enrich the content of the manuscript.
Thank you for your extremely kind and patient answers and changes. After expansion, you should take care of the fluidity of the narrative. The text is long, so it is very important to keep reader's attention
Answer: Thank you very much for your kind suggestions. We have carefully read the manuscript, modified the voice of various sentences in the text, and expounded the meaning of some experimental indicators in the manuscript, hoping to improve the fluency of the article and the readability of readers.
Correction of typos, missing or extra spaces and empty fragments are required.I hope that the next tests - according to the declarations will be perfect.
Answer: Thank you very much for your help with our manuscript. This is a mistake in our work. We reviewed the manuscript again according to your comments, corrected the spelling errors of words, deleted blank paragraphs according to the requirements of the instructions to authors, and revised the article format again.
Round 3
Reviewer 3 Report
Congratulations on your perseverance.
I hope that the next study will follow the solutions noted in the text to improve the methodology.